# Controversy of Peptide Cyclization from Tripeptide

**DOI:** 10.3390/molecules26020389

**Published:** 2021-01-13

**Authors:** Chung-Yin Lin, Subrata Chakraborty, Chia-Wei Wong, Dar-Fu Tai

**Affiliations:** 1Medical Imaging Research Center, Institute for Radiological Research, Chang Gung Memorial Hospital, Chang Gung University, Taoyuan 333423, Taiwan; 2Department of Nephrology and Clinical Poison Center, Chang Gung Memorial Hospital, Taoyuan 333423, Taiwan; 3Department of Chemistry, National Dong Hwa University, Hualien 974003, Taiwan; cpapu2000@yahoo.com (S.C.); linny5782@hotmail.com (C.-W.W.); 4Department of Life Science, National Dong Hwa University, Hualien 974003, Taiwan

**Keywords:** tripeptide, cyclic hexapeptide, solution phase synthesis, peptide cyclization

## Abstract

The present investigation reports an attempt to synthesize naturally occurring α-cyclic tripeptide *cyclo*(Gly-l-Pro-l-Glu) **1**, [*cyclo*(GPE)], previously isolated from the *Ruegeria* strain of bacteria with marine sponge *Suberites domuncula.* Three linear precursors, Boc-GPE(OBn)_2_, Boc-PE(OBn)G and Boc-E(OBn)GP, were synthesized using a solution phase peptide coupling protocol. Although *cyclo*(GPE) **1** was our original target, all precursors were dimerized and cyclized at 0 °C with high dilution to form corresponding α-cyclic hexapeptide, *cyclo*(GPE(OBn))_2_
**7**, which was then converted to cyclic hexapeptide *cyclo*(GPE)_2_
**2**. Cyclization at higher temperature induced racemization and gave cyclic tripeptide *cyclo*(GP_D_E(OBn)**) 9**. Structure characteristics of the newly synthesized cyclopeptides were determined using ^1^H-NMR, ^13^C-NMR and high-resolution mass spectrometry. The chemical shift values of carbonyls of **2** and **7** are larger than 170 ppm, indicating the formation of a cyclic hexapeptide.

## 1. Introduction

Marine sponge-derived natural cyclic peptides have generated much interest in recent years due to their privileged structures and persuasive biological activities (therapeutic potential). The surfaces and internal spaces of sponges provide a specific environmental niche that contains a high number of bacteria to exceed those of seawater by two or three orders of magnitude [1]. Moreover, several studies have increased interest in sponge-associated bacteria recently by proving that these bioactive compounds, when isolated from marine sponges, are symbiotically produced microorganisms [2]. Among these cyclic peptides, naturally occurring α-cyclic tripeptides and tetrapeptides are highly constrained because they have a small ring size [3]. Diverse biological activities of natural cyclic peptides, such as antimicrobial, cytotoxic, anti-HIV, nematicidal and anti-inflammatory characteristics [4,5,6,7], increases their importance as targets for synthetic chemists. However, many biologically active cyclic peptides have been isolated [8,9,10,11,12], but not fully characterized [13,14]. Another hurdle is to meet the supply problem and target identification. Synthesis [15,16] is particularly useful for biological applications as well as to confirm the actual structure of natural products and target identification to overcome the hurdle.

It is extremely challenged for successful drug development of these often complex molecules. In many cases, total synthesis of natural products revealed structural incorrectness of the reported natural products [17,18,19]. Previously, we reported the synthesis and X-ray crystallographic characterization of marine cyclic tetrapeptide *cyclo*(Gly-l-Ser-l-Pro-l-Glu) [20]. Antitumor and antimicrobial activity of some cyclic tetrapeptides and tripeptides were examined [21]. In addition, some α-cyclic tripeptides also synthesized as the inhibitors for HMG-CoA reductase [22]. As a continuation of natural cyclopeptide synthesis, we attempted to synthesize a rare example of naturally occurring marine α-cyclic tripeptide, *cyclo*(Gly-l-Pro-l-Glu) **1** [8], for applications in vivo as it is more permeable and more resistant to degradation by digestive proteases [23]. When rendered into its linear form, Gly-l-Pro-l-Glu (GPE) is the *N*-terminal sequence of insulin-like growth factor (IGF-1) [24,25]. There is evidence that the peptide has neuroprotective properties and improves long-term function following brain injury or disease, including hypoxic-ischemic brain injury, chemical toxins and in animal models of Parkinson’s and Alzheimer’s disease [26,27,28]. Furthermore, GPE and GPE analogues are thus of interest as promising neuroprotective agents for the treatment of central nervous system (CNS) diseases [27,28]. Despite having therapeutic significance, several challenging features are still encountered in the synthesis, such as dimerization and racemization of linear natural amino acid sequence, cyclization only with all 3 amide linkages in *cis* configuration and less conformation to raise the accuracy of molecular docking due to its rigidity [22,29,30,31]. These factors can produce contamination to influence the in vivo ability of formulation/characterization for clinical applications.

To address these problems, this paper explores an attempt to synthesize the reported natural cyclic tripeptide **1** since it possesses a side-chain carbonyl group that may enable further functionalization after cyclization. Although the literature reported natural cyclic tripeptide, the characterization data of our synthesized compound turn out to be cyclic hexapeptide **2**. In addition, the cyclization of linear precursor at room temperature (RT) was involved with racemization to give a cyclic d-tripeptide **9.**

## 2. Results and Discussion

To synthesize the cyclic tripeptide, linear precursor Boc-G-P-E(OBn)_2_
**5** (Scheme 1) was synthesized using standard solution-phase peptide coupling protocols from Fmoc-l-proline. Subsequently, regioselective enzymatic hydrolysis of the α-benzyl ester on glutamate was achieved [32]. The linear precursor was then activated with pentafluorophenol and cyclized in pyridine at high dilution (2 mM, RT). The cyclization ended up with *cyclo*(GPE(OBn))_2_
**7**. Formation of dimer was confirmed by high-resolution mass spectrometry (Appendix A). Then, deprotection of the benzyl group furnished the synthesis of *cyclo*(GPE)_2_
**2**. The results indicate that GPE sequence is much easier to produce cyclic products, but others are probably more difficult to be cyclized. In contrast, Yudin et al. [33] used Boc-protected β-amino imide as a model substrate to demonstrate that medium-sized rings could be constructed through the collapse of cyclol intermediates derived from the intramolecular cyclization of β-amino imides upon heating to 50 °C for 4 h. We also monitored the cyclization step by using RP-HPLC to analyze the compounds. Pure *cyclo*(GPE)_2_OBn showed peak at retention time 3.78, and the reaction mixture showed one peak correspond to *cyclo*(GPE)_2_OBn and another was salt of TFA.

Analysis of spectroscopic data of reported natural product **1** and synthetic cyclic hexapeptide **2**, especially ^1^H and ^13^C-NMR data, creates a doubt about the correctness of the natural product structure. Table 1 shows the recorded carbonyl peaks for reported cyclic peptides in ^13^C-NMR. All the ring carbonyl peak values of reported cyclic tripeptides are lower than 170 ppm [34,35,36]. Carbonyl peak of proline in *cyclo*(l-Pro-l-BnG-d-Pro) [34], *cyclo*(Pro-Pro-BnG) [35], *cyclo*(l-Pro-l-Pro-d-Pro) [35] and *cyclo*(l-Pro-l-Pro-l-Pro) [36] showed peaks between δ 167 and 169 ppm. Glycine carbonyl [36], just like proline, also appeared in the range around δ 166 ppm.

In contrast, in the case of reported natural product **1**, ring carbonyl peak values for proline and glycine are much higher (δ 174.8 and 169.8 ppm) than for other reported cyclic tripeptides [34,35,36] and closer to our synthetic cyclic hexapeptide *cyclo*(GPE)_2_
**2** (δ 173.8 and 169.6 ppm). Table 2 shows complete comparison of ^13^C chemical shift values of *cyclo*(GPE) **1 [8]** and *cyclo*(GPE)_2_
**2**. To verify the spectra of cyclic monomer and dimer containing same amino acid sequence, we synthesized *cyclo* (serine-proline-glycine) (SPG) and *cyclo*(SPG)_2_. Compared with natural products, *cyclo*(SPG) containing l-serine instated l-glutamic acid in its amino acid sequence. Both the monomer and dimer showed completely different ^1^H and ^13^C-NMR spectra (unpublished work). ^13^C peak values of *cyclo*(SPG)_2_ are also quite similar to our synthesized hexapeptide *cyclo*(GPE)_2_
**2** (Figure 1). We used a 3:1 CD_3_OD/D_2_O ratio for NMR, which was the cause of little variation of NMR peaks.

Cyclization of fully deprotected linear G-P-E(OBn)-OH in DMF at various dilution conditions with different coupling reagent combinations such as PyBOP-HOBt, HATU-HOAt and BOP-HOBt also gave dimeric compound. After failure to acquire a monomeric compound, we prepared two other linear precursors, Boc-PE(OBn)G and Boc-E(OBn)GP. These two compounds were cyclized using the same procedure and producing the same dimer **2** with lower cyclization yield. We also tried microwave assisted cyclization of linear precursor with a series of coupling reagents in DMF as well as different activated ester compounds in pyridine with variation of temperature, power and dilution failed to give cyclic tripeptide **1**.

To identify the specific functional groups in our synthesized compound, an infrared spectrum was also useful to check the carbonyl position. Amide stretching of small cyclic peptides was around 1675 cm^−1^ and usually shifted to 1635 cm^−1^ for larger rings. Unfortunately, the reported *cyclo*(GPE) did not provide this information [8]. It is well known that cyclic tripeptide can be formed only when all peptide bonds are in crown form with *cis*-linkage [37,38]. Small ring size and crown structure will bring all the carbonyls closer and parallel to each other, which will increase the shielding effect of one carbonyl to another. As a result, carbonyls on the cyclic tripeptide ring shift upfield (lower than 170 ppm in ^13^C-NMR). Conversely, carbonyl peaks of cyclic hexapeptide usually appeared in the downfield region (higher than 170 ppm), because flexible cyclic hexapeptide contains both *cis* and *trans* links [39]. Their *cis trans* combination keeps ring carbonyls apart from each other, which results in a less shielding effect. Carbonyl peaks of reported natural product **1** along with our synthetic cyclic hexapeptide *cyclo*(GPE)_2_
**2** showed similar peak patterns to previously synthesized [40] and isolated cyclic hexapeptides [41,42,43].

The ^1^H-NMR of synthesized cyclic hexapeptide also showed a similar splitting pattern to reported natural product **1** in the same duterum solvent (Table 3). Only one set of GPE protons was observed for synthetic *cyclo*(GPE)_2_. All α-hydrogens of cyclic hexapeptide showed the same splitting (Gly dd, Pro dd, Glu dd) and coupling constants as reported natural product **1**. In addition to splitting, chemical shift values of α-hydrogens (Gly δ 4.33, 4.20, Pro δ 4.51, Glu δ 4.64) were also identical to the reported data. High-resolution mass spectra show a mass ion peak at 566.2349 corresponding to *cyclo*(GPE)_2_ and a peak at 283 corresponding to *cyclo*(GPE). In addition, our cyclic hexapeptide ([α]^28^_D_ −33 (*c* 0.1, MeOH)) also has similar optical rotation as reported ([α]^25^_D_ −27.8 (*c* 0.0010, MeOH)) [8]. These results demonstrated that reported natural product **1** is perhaps a cyclic hexapeptide.

As shown in Scheme 1, droping the α-pentafluorophenol ester of G-P-E(OBn)-OH to pyridine at 50 °C resulted in racemization to form *cyclo*(GPDE(OBn)) **9** (27%). Removal of the benzyl protecting group on glutamate generated a cyclic tripeptide *cyclo*(Gly-l-Pro-d-Glu) **10**. The analysis of ^13^C-NMR (Figure 2) and ESIMS data (284.1241 *m*/*z*) indicated that the synthesized cyclic tripeptide **10** was not closely related to reported natural product **1** or synthesized cyclic hexapeptide **2**. Their ^1^H-NMR data are shown in Table 4. As compared with the optical rotation, our synthesized cyclic tripeptide **10** has a distinct value ([α]^24^_D_ −150 (*c* 0.01, MeOH)), which is different to reported natural product **1** ([α]^25^_D_ −27.8 (*c* 0.0010, MeOH)) or synthesized compound 2 ([α]^28^_D_ −33 (*c* 0.1, MeOH)). Our results suggest that this cyclic tripeptide is *cyclo*(Gly-l-Pro-d-Glu), *cyclo*(GP_D_E) **10,** not *cyclo*(Gly-l-Pro-l-Glu) **1**. These results also point out that GPE sequence is much easier to cycle than other precursors.

The antibiotic activity of natural product 1 was reported [8]. To date, we have not found any cyclic tripeptide with antibiotic activity [21,22]. We have shown that the structure–activity relationship cannot be neglected. The size of the cyclic peptide is very critical. Previous studies have disclosed the importance of ring size between 4 and 14 residues on the antimicrobial activity [44,45], such as cyclic hexapeptides [40,46,47,48].

## 3. Materials and Methods

All reactions involving air or moisture sensitive reagents were carried out under a dry argon or nitrogen atmosphere using freshly distilled solvents. Pyridine and dioxane were distilled from CaH_2_. The NMR spectra were recorded on Bruker DRX 400 (^1^H at 400.13 MHz, and ^13^C at 100.03 MHz). Chemical shifts (δ) are reported in ppm relatively to the residual solvent peak (CHCl_3_, δ = 7.26, ^13^CDCl_3_, δ = 77.0; CD_2_HOD, δ = 3.34, ^13^CD_3_OD, δ = 49.0; D_2_O, δ = 4.47. MALDI TOF was performed on a Bruker Autoflex MALDI-TOF mass spectrometer (Bruker Daltonics, Breman, Germany). High-resolution electrospray ionization mass spectrometry (ESI-MS) was performed on a Shimadzu-LCMS-IT-TOF mass spectrometer (Shumadzu, Kyoto, Japan). Infrared spectra were recorded on a Perkin Elmer Spectrum one FT-IR spectrometer (PerkinElmer, Shelton, CT, USA) using KBr pellets (4000–400 cm^−1^).

### 3.1. Fmoc-l-Prolyl-l-Glutamyl Dibenzyl Ester (**3**)

To a solution of dibenzyl L-glutamyl *p*-toluenesulphonate (8.88 g, 17.79 mmol) and triethylamine (3 g, 29.65 mmol) in dichloromethane (20 mL) in 50 mL round bottom flask Fmoc-l-proline (4 g, 11.86 mmol), DCC (2447 mg, 11.86 mmol) and HOBt (1601 mg, 11.86 mmol) was added at 0 °C. The reaction mixture was stirred for 2 h at 0 °C and 17 h at room temperature. The reaction was monitored by TLC and the resultant white mixture was filtered to remove 1,3-dicyclohexylurea (DCU). After evaporation of the filtrate, the resultant mass was dissolved in ethyl acetate and washed successively with 5% citric acid solution, water, 5% sodium bicarbonate solution and water. The organic extract was concentrated and purified by column chromatography using hexane/ethyl acetate (70:30) as eluent to obtain compound **3** as a white solid (6 g, 78%). [α]^22^_D_ = −56 (*c* 0.1, CHCl_3_). ^1^H-NMR (CDCl_3_, 400 MHz) δ 1.88–1.99 (m, 4H), 2.25–2.40 (m, 4H), 3.46–3.56 (m, 2H), 4.24–4.34 (m, 3H), 4.41–4.46 (m, 2H), 4.61–4.69 (m, 1H), 5.04–5.17 (m, 4H), 7.30–7.39 (m, 14H), 7.59 (s, 2H), 7.75–7.77 (d, 2H, *J* = 7.16 Hz). ^13^C-NMR (CDCl_3_, 100 MHz) δ 27.2, 28.5, 30.1, 34.0, 47.1, 47.3, 51.8, 60.4, 66.5, 67.3, 67.7, 120.0, 125.1, 125.2, 127.1, 127.7, 128.2, 128.3, 128.5, 128.5, 128.6, 135.3, 135.8, 143.8, 144.0, 171.4, 171.7, 172.5. IR (KBr) 3326, 3065, 2952, 1736, 1704, 1532, 1451, 1418, 1353, 1262, 1169, 1119, 1089, 988, 758, 740, 698, 621 cm^−1^. MALDI TOF: *m*/*z* calcd. for C_39_H_38_N_2_O_7_Na [M + Na]^+^ 669.257; found 669.170.

### 3.2. l-Prolyl-l-Glutamyl Dibenzyl Ester (**4**)

To a solution of Fmoc-l-prolyl-l-glutamyl dibenzyl ester **3** (6 g, 9.28 mmol) in dichloromethane (16 mL) in a round bottom flask piperidine was added (4 mL). The mixture was stirred for 2 h. The reaction was monitored by TLC which showed the complete disappearance of starting material. The reaction mixture was concentrated and purified by column chromatography using DCM/MeOH (97:3) as eluent to obtain compound **4** as a pale yellow liquid (3.7 g, 94%). ^1^H-NMR (CDCl_3_, 400 MHz) δ 1.65–1.70 (m, 4H), 1.85–1.88 (m, 1H), 2.00–2.10 (m, 2H), 2.29–2.41 (m, 3H), 2.87–2.99 (m, 2H), 3.69–3.73 (m, 1H), 4.62–4.64 (m, 1H), 5.00–5.14 (m, 4H), 7.32 (s, 10H), 8.13–8.15 (d, 1H, *J* = 8.48 Hz). ^13^C-NMR (CDCl_3_, 100 MHz) δ 26.2, 27.6, 30.3, 30.9, 47.3, 51.1, 60.5, 66.5, 67.2, 128.2, 128.3, 128.4, 128.6, 128.6, 135.3, 135.8, 171.7, 172.3, 175.5. MALDI TOF: *m*/*z* calcd. for C_24_H_29_N_2_O_5_ [M + H]^+^ 425.207; found 425.267.

### 3.3. Boc-Glycyl-l-Prolyl-l-Glutamyl Dibenzyl Ester (**5**)

Dichloromethane (20 mL) and triethylamine (953 mg, 9.43 mmol) were taken in a round bottom flask and cooled at 0 °C. To this, l-prolyl-l-glutamyl dibenzyl ester **4** (2 g, 4.71 mmol) was added and stirred for 5 min. Boc-glycine (825 mg, 4.71 mmol), DCC (972 mg, 4.71 mmol) and HOBt (636 mg, 4.71 mmol) were added. The reaction mixture was stirred 2 h at 0 °C and 17 h at rt. The reaction was monitored by TLC which showed the complete disappearance of both starting materials. The resultant white mixture was filtered to remove 1,3-dicyclohexyleurea (DCU). Filtrate was dried, and residue was dissolved in ethyl acetate. The organic layer was washed with 5% aq. citric acid solution followed by water and then with 5% aq. NaHCO_3_ solution followed by water. The organic layer was dried over Na_2_SO_4_, and the mixture was purified by column chromatography using hexane/ethyl acetate (65:35) as eluent to obtain compound **5** as colorless liquid. (2.25 g, 82%). [α]^22^_D_ = −66 (*c* 0.1, CHCl_3_). ^1^H-NMR (CDCl_3_, 400 MHz) δ 1.4 (s, 9H), 1.95–2.06 (m, 4H), 1.97–2.04 (m, 2H), 2.17–2.24 (m, 2H), 2.36–2.41 (m, 2H), 3.31–3.37 (m, 1H), 3.44–3.49 (m, 1H), 3.82 (dd, 1H, *J* = 3.92 Hz, *J* = 17.2 Hz), 3.96 (dd, 1H, *J* = 4.5 Hz, *J* = 17.4 Hz), 4.49–4.53 (m, 1H), 4.54–4.58 (m, 1H), 5.06–5.16 (m, 4H), 5.25–5.26 (d, 1H, *J* = 1.76 Hz), 5.47 (s, 1H), 7.26–7.38 (m, 10H). ^13^C-NMR (CDCl_3_, 100 MHz) δ 24.8, 26.9, 27.9, 28.4, 30.2, 43.1, 46.3, 51.9, 60.1, 66.5, 67.2, 79.7, 135.3, 135.8, 155.8, 168.5, 171.2, 171.4, 172.7. IR (KBr) 3316, 3034, 2977, 1733, 1653, 1528, 1455, 1391, 1367, 1251, 1168, 1057, 1029, 971, 866, 751, 699, 581 cm^−1^. HRMS (ESI): *m*/*z* calcd. for C_31_H_40_N_3_O_8_ [M + H]^+^ 582.2810; found 582.2814.

### 3.4. Boc-Glycyl-l-Prolyl-l-Glutamic Acid γ-Benzyl Ester (**6**)

To a solution of 0.1 M, pH 7 phosphate buffer (32 mL) containing B. *subtilis* protease (Sigma type–VIII) (10 mg) in a 100 mL round bottom flask a solution of Boc-glycyl-l-prolyl-l-glutamyl dibenzyl ester **5** (1 g, 1.72 mmol) in acetone was added drop by drop (8 mL). The reaction mixture was stirred at 35 °C overnight. The reaction was monitored by TLC which showed the complete disappearance of the starting material. Then, the pH of the solution was basified to 8 and unchanged ester was extracted with ethyl acetate. The aqueous layer was acidified to pH 2, centrifuged to remove enzyme and extracted with ethyl acetate. The resultant organic layer was dried over Na_2_SO_4_, concentrated and purified by column chromatography using DCM/MeOH (97:3) as eluent to obtain compound **6** as a white solid. (727 mg, 86%). [α]^22^_D_ = −64 (*c* 0.1, CHCl_3_). ^1^H-NMR (CDCl_3_, 400 MHz) δ 1.41 (s, 9H), 1.88–1.93 (m, 2H), 2.16–2.25 (m, 2H), 2.37–2.50 (m, 2H), 3.39–3.41 (m, 1H), 3.51–3.55 (m, 1H), 3.84 (dd, 1H, *J* = 3.84 Hz, *J* = 17.28 Hz), 4.00 (dd, 1H, *J* = 5.64 Hz, *J* = 17.48 Hz), 4.48–4.53 (m, 2H), 5.07 (s, 2H), 5.65 (s, 1H), 7.28–7.33 (m, 5H), 7.46–7.48 (d, 1H, *J* = 7.56 Hz), 8.20 (br s, 1H). ^13^C-NMR (CDCl_3_, 100 MHz) δ 24.8, 26.8, 28.3, 30.3, 43.0, 46.5, 51.9, 53.5, 60.4, 66.5, 79.9, 128.2, 128.3, 128.6, 135.6, 135.8, 156.1, 169.2, 171.6, 173.2, 173.6. IR (KBr) 3327, 2978, 1731, 1652, 1531, 1454, 1392, 1367, 1252, 1167, 1057, 865, 749, 699 cm^−1^. HRMS (ESI): *m*/*z* calcd. for C_24_H_33_N_3_O_8_ [M]^+^ 491.2268; found 491.2268.

### 3.5. Cyclo(Glycyl-l-Prolyl-l-Glutamyl(OBn))_2_ (**7**)

Boc-glycyl-l-prolyl-l-glutamyl γ-benzyl ester α-acid **6** (491.53 mg, 1 mmol) was added to a 10 (mL) round bottom flask containing DCM (2 mL) and cooled at 0 °C. Pentafluorophenol (202 mg, 1.1 mmol) and DCC (206 mg, 1 mmol) were added. The reaction mixture was stirred for 1 h at 0 °C and 22 h at rt. The solvent was evaporated, and the residue was dissolved in ethyl acetate and filtered to remove 1,3 dicyclohexyleurea (DCU). The filtrate was evaporated to obtain Boc-glycyl-l-prolyl-l-glutamyl γ-benzyl ester α-pentafluorophenol ester. Boc-glycyl-l-prolyl-l-glutamyl γ-benzyl ester α-pentafluorophenol ester (658 mg, 1 mmol) was taken in a (25 mL) round bottom flask and cooled to 0 °C. TFA/DCM (1:1) (4 mL) was added and stirred 1 h at 0 °C. The solvents were evaporated and triturated with ether followed by decanting to remove free pentafluorophenol. The residue was dried in vacuum, which was used directly in the ensuing cyclization procedure. Glycyl-l-prolyl-l-glutamyl γ-benzyl ester α-pentafluorophenol ester (492 mg, 1 mmol) was dissolved in dioxane (20 mL). This solution was added dropwise with efficient stirring to a 1 L round bottom flask containing 500 mL pyridine. Addition was completed after 6 h. After stirring over 36 h at rt, the solvent was distilled off. Residue was dissolved in DCM. The organic layer was washed with 1 N HCl solution, followed by water and then with 5% sodium bicarbonate solution, followed by water. The organic layer was dried over sodium sulphate, and the mixture was purified by column chromatography using DCM/MeOH (95:5) as eluent to obtain compound **7** as a white solid. (110 mg, 30%). [α]^22^_D_ = −46 (*c* 0.5, CHCl_3_). ^1^H-NMR (CDCl_3_, 400 MHz) δ 1.93–1.96 (m, 8H), 2.22–2.29 (m, 4H), 2.38–2.45 (m, 4H), 3.44–3.46 (m, 2H), 3.75–3.81 (m, 4H), 4.20–4.21 (d, 1H, *J* = 4.72 Hz), 4.24–4.25 (d, 1H, *J* = 4.96 Hz), 4.33 (dd, 2H, *J* = 4.04 Hz), 4.44–4.49 (m, 2H), 5.07 (s, 4H), 7.13–7.15 (d, 2H, *J* = 8.24 Hz), 7.29–7.34 (m, 10H), 7.73 (s, 2H). ^13^C-NMR (CDCl_3_, 100 MHz) δ 24.9, 26.9, 29.7, 30.6, 42.9, 46.9, 52.1, 61.4, 66.5, 128.2, 128.3, 128.4, 128.6, 135.8, 168.2, 171.5, 171.8, 173.2. IR (KBr) 3329, 3065, 2951, 1732, 1647, 1516, 1454, 1386, 1325, 1260, 1165, 1004, 738, 699, 597 cm^−1^. ESI: *m*/*z* calcd. for C_38_H_47_N_6_O_10_ [M + H]^+^ 747.33; found 747.27.

### 3.6. Cyclo(Glycyl-l-Prolyl-l-Glutamyl)_2_ (**2**)

*Cyclo*(glycyl-l-prolyl-l-glutamyl(OBn))_2_**7** (200 mg, 0.27 mmol) was dissolved in 4 mL of MeOH in a 10 mL round bottom flask and 20% palladium hydroxide on carbon was added under nitrogen. The vessel was puged three times with nitrogen and three times with hydrogen, and the reaction mixture was then stirred for 2 h under hydrogen at atmospheric pressure. The catalyst was removed by filtration, and the filtrate was evaporated to obtain compound **2** as a white solid. (144 mg, 95%). [α]^28^_D_ = −33 (*c* 0.1, MeOH). ^1^H-NMR (CD_3_OD, D_2_O, 400 MHz) δ 2.13–2.21 (m, 6H), 2.27–2.38 (m, 6H), 2.41–2.53 (m, 4H), 3.71–3.77 (m, 2H), 3.94–3.99 (m, 2H), 4.2 (d, 2H, *J* = 17.32 Hz), 4.33 (d, 2H, *J* = 17.56 Hz), 4.51 (dd, 2H, *J* = 8.72 Hz, *J* = 6.12 Hz), 4.64 (dd, 2H, *J* = 10.28 Hz, *J* = 4.08 Hz). ^13^C-NMR (CD_3_OD, D_2_O, 400 MHz) δ 26.1, 29.0, 30.7, 35.4, 43.7, 47.7, 54.5, 63.5, 169.6, 173.8, 175.0, 181.6. IR (KBr) 3405, 2944, 1635, 1564, 1404, 1328, 1189, 1111, 1050, 919, 599 cm^−1^. HRMS (ESI): *m*/*z* calcd. for C_24_H_34_N_6_O_10_ [M]^+^ 566.2336; found 566.2349.

### 3.7. Cyclo(Glycyl-l-Prolyl-d-Glutamyl) (**10**)

Boc-glycyl-l-prolyl-l-glutamyl γ-benzyl ester α-acid **6** (100 mg, 0.2 mmol) was added to a 10 (mL) round bottom flask containing DCM (2 mL), pentafluorophenol (41.5 mg, 0.22 mmol) and DCC (41.3 mg, 0.3 mmol). Reaction mixture was stirred for 1.5 h at rt. The solvent was evaporated, and the residue was dissolved in ethyl acetate and filtered to remove 1,3 dicyclohexyleurea (DCU). The filtrate was evaporated to obtain Boc-glycyl-l-prolyl-l-glutamyl γ-benzyl ester α-pentafluorophenol ester. Boc-glycyl-l-prolyl-l-glutamyl γ-benzyl ester α-pentafluorophenol ester (131.5 mg, 0.2 mmol) was taken in TFA/DCM (1:1) (2 mL) and stirred for 1 h at 0 °C. The solvents were evaporated to remove TFA. The residue was dried in vacuum, which was used directly in the ensuing cyclization procedure. The TFA salt of glycyl-l-prolyl-l-glutamyl γ-benzyl ester α-pentafluorophenol ester (111.5 mg, 0.2 mmol) was dissolved in 1,4-dioxane. This solution was added dropwise with efficient stirring to a 1 L round bottom flask containing 500 mL pyridine at 50 °C. After dropping was completed, the temperature was lowered to rt and stirring over 60 h at rt, the solvent was distilled off. The residue mass was dissolved in DCM, hexane, ethyl ester and washed with ethyl acetate to obtain *cyclo*(glycyl-l-prolyl-d-glutamyl(OBn)) (20 mg, 27%).

*Cyclo*(glycyl-l-prolyl-d-glutamyl(OBn)) (20 mg, 0.053 mmol) was dissolved in 2 mL MeOH and DMF in a 10 mL round bottom flask, and palladium hydroxide on carbon was added under nitrogen. The reaction mixture was then stirred for 2 h. The catalyst was removed by filtration, and the filtrate was evaporated to obtain a white solid compound (10.5 mg, 70%). [α]^24^_D_ = −150 (c 0.01, MeOH). ^1^H-NMR (D_2_O, 400 MHz) δ 2.06–2.18 (m, 3H), 2.19–2.24 (m, 2H), 2.32–2.42 (m, 2H), 2.64–2.70 (m, 1H), 3.35–3.59 (m, 1H), 3.67–3.72 (m, 1H), 3.79 (t, 1H, *J* = 6.4 Hz), 4.10 (dd, 1H, *J* = 17.1 Hz, 3.3 Hz), 4.17 (t, dd, 1H, *J* = 4.6 Hz), 4.24 (dd, 1H, *J* = 17.4 Hz, 7.6 Hz). ^13^C-NMR (D_2_O, 100 MHz) δ 22.1, 26.4, 27.8, 32.0, 45.0, 45.6, 57.2, 57.7, 161.4, 164.9, 174.4. IR (KBr) 3375, 2927, 2328, 2858, 1652, 1430, 1314, 1151, 1097, 985, 874, 751, 631 cm^−1^. HRMS(ESI): *m*/*z* calcd. for C_12_H_18_N_3_O_5_ [M + H]^+^ 284.1242; found 284.1242.

## 4. Conclusions

In summary, an α-cyclic hexapeptide *cyclo*(GPE)_2_
**2** was synthesized via cyclization of fully deprotected linear precursor with a series of coupling reagents in DMF. Additionally, we were able to obtain *cyclo*(GPDE) with much synthetic effort. Even though the characterization data of the reported natural product and our synthetic *cyclo*(GPE)_2_ indicate that the reported compound may be a dimer of *cyclo*(GPE), this strategy still provided a way to synthesize cyclic peptide of varying ring size. In light of the cyclic peptide synthesis using a solution phase method, it can also provide opportunities to explore the synthesis of cyclic products with various functionalities. The unusual antibiotic activity of natural product 1 raised our suspicion as a cyclic tripeptide. The characterization data of the reported natural product and our synthetic *cyclo*(GPE)_2_ also indicate that the reported compound may be a dimer of *cyclo*(GPE).

## Data Availability

The data presented in this study are availiable on request from corresponding authors. The data are not publicly available due to confidentiality due diligence.

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
