# Peer review of "Controversy of Peptide Cyclization from Tripeptide"

_molecules, 2021, doi:10.3390/molecules26020389_

Round 1
Reviewer 1 Report
This paper had attempted to synthesize cyclic tripeptide cyclo (GPE) but synthesized cyclic hexapeptide cyclo (GPE)2.
This result is not publishable at this stage. The author should figure out an optimized reaction condition to synthesize cyclic (GPE); otherwise, they should demonstrate cyclo (GPE)2 has better or similar activity to cyclo (GPE).
Author Response
We greatly appreciate the reviewer’ comments guiding us to revise the manuscript accordingly. The contents and the clarity of our paper are much improved in the revised version. Point-by-point responses are listed following every specific the comment in this response letter. The revised parts of the manuscript (our responses) are represented in red, bold and underline in the text.
Reviewer #1:
Comments: This result is not publishable at this stage. The author should figure out an optimized reaction condition to synthesize cyclic (GPE); otherwise, they should demonstrate cyclo (GPE)2 has better or similar activity to cyclo (GPE).
Response: We appreciate the reviewer’s suggestion. Our attempts explored the synthesis of the natural cyclic tripeptide and its GPE analogues. Indeed, the result’s survey revealed striking differences to previous study. Besides, the side-chain carbonyl group enables further functionalization after cyclization. We carried out cyclo (GPDE) and cyclo (GPE)2 experiments to demonstrate a way to synthesize cyclic peptide of varying ring size. To further confirm the spectra of cyclic monomer and dimer containing same amino acid sequence, we also prepared cyclo (serine-proline-glycine) (SPG) and cyclo(SPG)2 to verify different 1H and 13C NMR spectra. In contrast to the relativity of cyclic (GPE), this tendency to publish our synthesized compound turns out to be cyclic hexapeptide or cyclic D-tripeptide is so anchored in scientific journals that even studies that contradict to a previous study with are seen and are not even reviewed.
Peptide cyclization generally improves the selective binding, uptake, potency and stability of linear precursors [Ref. 1, 21, 22]. Therefore, cyclic peptides should process similar activity to linear peptides [14, 15, 20-22]. Therefore, it is worthy to further study the effect of the neuroprotection of the brain injury using cyclo (GPE)2 at longer time points post-treatments in vivo and in vitro.
We added this paragraph to Results and Discussion Section: “In addition, the antibiotic activity of natural product 1 was reported [8]. Therefore, the structure-activity relationship cannot be neglected. However, we didn’t find any cyclic tripeptide with antibiotic activity [21, 22]. So is other. In contrast, many studies have disclosed the importance of ring size between 4 to 14 residues on the antimicrobial activity [44, 45]. Cyclic hexpeptides have especially been known to possess antibiotic activity [46-48]. These evidences point out the controversy of natural product 1 as a cyclic tripeptide.”
Reference in this response:
Ref. 1. Roxin, A., Zheng, G. Flexible or fixed: a comparative review of linear and cyclic cancer-targeting peptides. Future Med. Chem. 2012, 4, 1601-1618.
Ref. 8. Mitova, M.; Popov, S.; De Rosa, S. Cyclic peptides from a Ruegeria strain of bacteria associated with the sponge Suberites domuncula. J. Nat. Prod. 2004, 67, 1178–1181.
Ref. 14. Davies, J. S. The cyclization of peptides and depsipeptides. J. Peptide Sci. 2003, 9, 471-501.
Ref. 15. Hamada, Y.; Shioiri, T. Recent progress of the synthetic studies of biologically active marine cyclic peptides and depsipeptides. Chem. Rev. 2005, 105, 4441–4482.
Ref. 20. Chakraborty, S.; Tyagi, P.; Tai, D. F.; Lee, G. H.; Peng, S. M. A lead (II) 3D coordination polymer based on a marine cyclic peptide motif. Molecules 2013, 18, 4972–4985.
Ref. 21. Chakraborty, S.; Tai, D. F.; Lin, Y. C.; Chiou, T. W. Antitumor and antimicrobial activity of some cyclic tetrapeptides and tripeptides derived from marine bacteria. Mar. Drugs 2015, 13, 3029-3045.
Ref. 22. Chakraborty, S.; Lin, S. H.; Shiuan, D.; Tai, D. F. Syntheses of some a-cyclic tripeptides as potential inhibitors for HMG-CoA reductase. Amino Acids 2015, 47, 1495-1505.
Ref. 44. Lee, D.L.; Hodges, R.S. Structure-activity relationships of de novo designed cyclic antimicrobial peptides based on gramicidin S. Biopolymers 2003, 71, 28–48.
Ref. 45. Falanga, A.; Nigro, E.; De Biasi, M. G.; Daniele, A.; Morelli, G.; Galdiero, S.; Scudiero, O. Cyclic peptides as novel therapeutic microbicides: engineering of human defensin mimetics, Molecules 2017, 22, 1217.
Ref. 46. Dartois, V.; Sanchez-Quesada, J.; Cabezas, E.; Chi, E.; Dubbelde, C.; Dunn, C.; Granja, J.; Gritzen, C.; Weinberger, D.; Reza Ghadiri, M.; Parr, T. R. Jr. Systemic Antibacterial activity of novel synthetic cyclic peptides. Antimicrob. Agents Chemother. 2005, 49, 3302–3310.
Ref. 47. Scheinpflug, K.; Krylova, O.; Nikolenko, H.; Thurm, C.; Dathe, M. Evidence for a novel mechanism of antimicrobial action of a cyclic R-,W-rich hexapeptide. PLoS ONE 2015, 10, e0125056.
Ref. 48. Wiese, J.; Abdelmohsen, U. R.; Motiei, A.; Humeida, U. H.; Imhoff, J. F. Bacicyclin, a new antibacterial cyclic hexapeptide from Bacillus sp. strain BC028 isolated from Mytilus edulis. Bioorg Med. Chem. Lett. 2018, 28, 558-561.

Reviewer 2 Report
The submitted manuscript from Tai and co-workers aims at synthesizing the naturally-occurring cyclic tripeptide cyclo-(Gly-L-Pro-L-Glu), isolated in 2004 from a Ruegeria strain of bacteria associated with the sponge Suberites domuncula. The authors describe the attempt of achieving the head-to-tail cyclization from several linear precursors as well as the utilization of dissimilar peptide coupling reagents, without success. Instead, they report the obtention of the cyclic hexapeptide cyclo-(Gly-L-Pro-L-Glu)2, produced from an initial dimerization reaction with subsequent ring closure. Interestingly, the obtained cyclic hexapeptide shares several similarities with the previously isolated natural product cyclo-(Gly-L-Pro-L-Glu), as confirmed by a comparison of NMR and optical rotation data, which points out to an inconsistency in the original paper where the structure of the natural product is described. Furthermore, the authors synthesize two additional cyclic peptides: cyclic tripeptide cyclo-(Ser-Pro-Gly) and the dimeric analog cyclo-(Ser-Pro-Gly)2, proving remarkable differences in the 13C chemical shifts of tri- and hexapeptides to be also observed among the natural and here-synthesized cyclic peptides. Altogether, the paper concludes that the synthesized cyclo-(Gly-L-Pro-L-Glu)2 should be indeed the correct structure of the natural product.
On a general basis, the manuscript is written concisely, and the main objectives and problematics are properly addressed. The total synthesis of natural products is highly relevant, and particularly cyclic peptides from marine organisms are promising molecules with potential biomedical applications. Particularly, the synthesis of cyclic tripeptides is highly challenging, since the conformational requirements associated with the ring-closing step conducts quite frequently to the formation of undesired side reactions, including racemization and oligomerization. The introduction describes the importance of the work and tracks an appropriate amount of references. The synthetic challenge associated with the synthesis of cyclic tripeptides could have been described in more detail, and a citation to A. Yudin Nat. Chem. 2011, 3, 509–524 should be added.
Together with the considerable synthetic effort carried out to attempt the synthesis of the cyclic tripeptide, the main achievement of the manuscript is to bring attention to the possibility that the original paper describing the isolation of the cyclic tripeptide cyclo-(Gly-L-Pro-L-Glu) could have reported a false structure (i.e the dimer cyclo-(Gly-L-Pro-L-Glu)2). Nevertheless, it should be noticed that:
- The HRMS(ESI) data reports the expected molecular ion at m/z 566.2349 ([M]+, calculated 566.2336). ESI-MS ionization techniques of small to medium peptides usually deliver pseudo-molecular ions (commonly [M+H]+ or [M+Na]+) and molecular ions are generally not observed. Furthermore, the authors describe the observation of a peak at half of that mass (283 described by the authors, calculated 283.1168). This data does not unequivocally point out to the formation of the dimer, since the peak at 566 could be the bimolecular adduct of the tripeptide ([2M]+ according to the data shown, usually and much more common [2M+H]+). To unequivocally distinguish by HRMS the presence or not of the dimer, the authors muss perform MS/MS experiments. In this way, the fragmentation of the molecular (or pseudo molecular) ion, when belonging to the dimer (hexapeptide), will lead to ions of intermediate m/z value (i.e. lower than 566 and higher than 283, belonging to the pentapeptidic and tetrapeptidic ions). Additionally, the isotopic pattern of the peak at 283 could confirm if this peak belongs to the mono- or bi-charged species. None of this information is discussed in the manuscript.
- The NMR comparison between synthesized and literature-reported peptides shows clear similarities, suggesting that they are the same compound. Nevertheless, there is evidence in the literature that NMR techniques cannot always distinguish between small cyclic peptides and their dimers (See, e.g. Wessjohann and co-workers, Lett. 2017, 19, 15, 4022–4025), and should be used carefully to perform such analysis since spin systems in monomers and dimers are the same. More important, a paper from Yudin and co-workers (Chem. Eur. J. 2017, 23, 13319 – 13322, see Supp. information) describes the calculation of the chemical shifts for the reported natural cyclic tripeptide, showing a good correlation between calculated and observed chemical shifts. This paper also reinforces the originally proposed tripeptidic structure and must be cited and commented by the authors since it directly contradicts the results here described.
- A literature-based analysis of 13C data reveals that rigid rings from cyclic tripeptides show relatively shielded carbonyl chemical shifts (below 170 ppm), which contrasts with the relatively higher values reported for the natural product. It should be noticed that most of the reviewed data is reported in non-polar solvents while the synthetic peptide is recorded in D2O/MeOD, this should be clearly illustrated in the text. The authors justify this tendency by stating that “ cyclic tripeptide can be formed only when all peptide bonds are in crown form with cis-linkage”. Even when it is true that the crown all-s-cis configuration is highly populated in cyclic tripeptides, this is reported as the exclusive conformation for cyclo-(L-Pro-L-Pro-L-Pro). Other model cyclic peptides (e.g. cyclo-(Gly-Gly-Gly) show similar population of trans-cis-cis. For the synthetic cyclo-(Gly-L-Pro-L-Glu)2), bidimensional spectra should be provided to guarantee the correct assignment of spin systems and carbonyl carbons (COSY, HSQC, HMBC, NOESY/ROESY) and the identification of key interresidual NOE contacts which contribute in determining the s-cis or s-trans arrangement of some of the amino acids (such an analysis is also described in the original paper, where s-trans conformation for the Gly-Pro linkage was identified for the isolated cyclo-(Gly-L-Pro-L-Glu)).
The lower point of the present manuscript is the lack of experimental information which guarantees the reproducibility of the described work. The authors should provide supplementary material showing all the recorded spectra. Chromatograms of the pure cyclic peptides and the gradients utilized should be provided (there are some references to HPLC analysis during the manuscript but no data is shown), as well as the retention times. The method utilized for the purification of the peptides is not described in the experimental section, neither their Rf nor the purity of the final cyclic peptides. Furthermore, the authors claim to have synthesized the cyclic tripeptide cyclo-(Ser-Pro-Gly) and its dimer cyclo-(Ser-Pro-Gly)2, but no description of the synthesis is shown. This latter should be contained in the final manuscript.
Finally, it should be pointed out that the quality of the English language should be improved. To some extend, ideas are not always clearly exposed and the reader faces difficulties understanding some of the statements. Sometimes prepositions and articles are omitted, reason why the revision of the language is recommended.
Author Response
We greatly appreciate the reviewer’ comments guiding us to revise the manuscript accordingly. The contents and the clarity of our paper are much improved in the revised version. Point-by-point responses are listed following every specific the comment in this response letter. The revised parts of the manuscript (our responses) are represented in bold and underline in the text.
Reviewer #2:
Comments: On a general basis, the manuscript is written concisely, and the main objectives and problematics are properly addressed. The total synthesis of natural products is highly relevant, and particularly cyclic peptides from marine organisms are promising molecules with potential biomedical applications. Particularly, the synthesis of cyclic tripeptides is highly challenging, since the conformational requirements associated with the ring-closing step conducts quite frequently to the formation of undesired side reactions, including racemization and oligomerization. The introduction describes the importance of the work and tracks an appropriate amount of references. The synthetic challenge associated with the synthesis of cyclic tripeptides could have been described in more detail, and a citation to A. Yudin Nat. Chem. 2011, 3, 509–524 should be added.
Response:
We thank the reviewer’s comments. Additional descriptions and references have been added in the revised version to describe the synthetic challenge associated with the synthesis of cyclic tripeptides in the “Introduction” per reviewer’s suggested.
“Although having therapeutic significance, several challenging features are still encountered in the synthesis, such as dimerization and racemization of linear natural amino acid sequence, cyclization only with all 3 amide linkages in cis configuration, and less conformation to raise the accuracy of molecular docking due to its rigidity [22, 29-31]. These factors can produce contamination to influence the vivo ability of formulation/ characterization for clinical applications.
To address these problems, this paper explores an attempt …….”
Reference in this response:
Ref. 22. Chakraborty, S.; Lin, S. H.; Shiuan, D.; Tai, D. F. Syntheses of some a-cyclic tripeptides as potential inhibitors for HMG-CoA reductase. Amino Acids 2015, 47, 1495-1505.
Ref. 29. White, C. J.; Yudin, A. K. Contemporary strategies for peptide macrocyclization. Nat. Chem. 2011, 3, 509-524.
Ref. 30. Kartha, G.; Ambady, G.; Shankar, P. V. Structure and conformation of a cyclic tripeptide. Nature 1974, 247, 204-205.
Ref. 31. Meng, X. Y.; Zhang, H. X.; Mezei, M.; Cui, M. Molecular docking: a powerful approach for structure-based drug discovery. Curr. Comput. Aided Drug Des. 2011, 7, 146-157.
C1: The HRMS(ESI) data reports the expected molecular ion at m/z 566.2349 ([M]+, calculated 566.2336). ESI-MS ionization techniques of small to medium peptides usually deliver pseudo-molecular ions (commonly [M+H]+ or [M+Na]+) and molecular ions are generally not observed. Furthermore, the authors describe the observation of a peak at half of that mass (283 described by the authors, calculated 283.1168). This data does not unequivocally point out to the formation of the dimer, since the peak at 566 could be the bimolecular adduct of the tripeptide ([2M]+ according to the data shown, usually and much more common [2M+H]+). To unequivocally distinguish by HRMS the presence or not of the dimer, the authors must perform MS/MS experiments. In this way, the fragmentation of the molecular (or pseudo molecular) ion, when belonging to the dimer (hexapeptide), will lead to ions of intermediate m/z value (i.e. lower than 566 and higher than 283, belonging to the pentapeptidic and tetrapeptidic ions). Additionally, the isotopic pattern of the peak at 283 could confirm if this peak belongs to the mono- or bi-charged species. None of this information is discussed in the manuscript.
R1: We respectively thank the reviewer’s suggestions. In fact, all synthesized compounds were completely characterized by NMR spectroscopy, FTIR, MALDI TOF mass spectrometry, and HRMS. New compounds were also given to identify selectively and effectively dimer structures. In addition, in the case of synthesized products, ring carbonyl peak values for proline, glycine and glutamate were previously ascertained [20-22]. These were used in biological activities and inhibitors and which was been validated according to Marine Drugs and Amino Acids journals, respectively. Therefore, the formation of the bimolecular adduct of other protein components were excluded as determined through spectroscopic data.
Reference in this response:
Ref. 20. Chakraborty, S.; Tyagi, P.; Tai, D. F.; Lee, G. H.; Peng, S. M. A lead (II) 3D coordination polymer based on a marine cyclic peptide motif. Molecules 2013, 18, 4972–4985.
Ref. 21. Chakraborty, S.; Tai, D. F.; Lin, Y. C.; Chiou, T. W. Antitumor and antimicrobial activity of some cyclic tetrapeptides and tripeptides derived from marine bacteria. Mar. Drugs 2015, 13, 3029- 3045.
Ref. 22. Chakraborty, S.; Lin, S. H.; Shiuan, D.; Tai, D. F. Syntheses of some a-cyclic tripeptides as potential inhibitors for HMG-CoA reductase. Amino Acids 2015, 47, 1495-1505.
C2: The NMR comparison between synthesized and literature-reported peptides shows clear similarities, suggesting that they are the same compound. Nevertheless, there is evidence in the literature that NMR techniques cannot always distinguish between small cyclic peptides and their dimers (See, e.g. Wessjohann and co-workers, Lett. 2017, 19, 15, 4022–4025), and should be used carefully to perform such analysis since spin systems in monomers and dimers are the same. More important, a paper from Yudin and co-workers (Chem. Eur. J. 2017, 23, 13319 – 13322, see Supp. information) describes the calculation of the chemical shifts for the reported natural cyclic tripeptide, showing a good correlation between calculated and observed chemical shifts. This paper also reinforces the originally proposed tripeptidic structure and must be cited and commented by the authors since it directly contradicts the results here described.
R2: We appreciate the reviewer’s suggestion. Although synthesis of cyclo (GPDE) and cyclo (GPE)2 were performed, cyclo (serine-proline-glycine) (SPG) and cyclo(SPG)2 were also synthesized to demonstrate the spectra of cyclic monomer and dimer containing similar amino acid sequence using different 1H and 13C NMR, mass, and FTIR spectrometry. These two sets of cyclotripeptides were checked respectively to their dimer structures.
In order to investigate the influence of naturally occurring a-cyclic tripeptide, we arranged three linear precursors Boc-GPE(OBn)2, Boc-PE(OBn)G and Boc-E(OBn)GP using solution-phase peptide coupling protocol in this study and dimerized and cyclized at low temperature with high dilution to form corresponding a-cyclic hexapeptide. By comparing the cyclization at higher temperature and gave cyclic tripeptide, we could identify the individual influence of temperature, solvent, dilution on the synthesis of small-sized cyclic peptides. In contrast, Yudin et al. [Ref. 33] used Boc-protected b-amino imide as a model substrate to demonstrate that medium-sized rings could be constructed to overcome the synthesized limitations, such as high entropic barriers, unwanted oligomerization, poor yields, and limited substrate accessibility. They showed that a robust methodology for the synthesis of challenging medium-sized rings could dramatically achieve through the collapse of cyclol intermediates derived from the intramolecular cyclization of b-amino imides upon heating to 50 °C for 4 h. This indicates that the configuration between small-sized cyclic peptides and medium-sized cyclic peptides in the presence of amide linkage is a major factor for varying ring size.
The following was added to Results and Discussion Section: “…… more difficult to be cyclized. In contrast, Yudin et al. [33] used Boc-protected b-amino imide as a model substrate to demonstrate that medium-sized rings could be constructed through the collapse of cyclol intermediates derived from the intramolecular cyclization of b-amino imides upon heating to 50 °C for 4 h.”
Reference in this response:
Ref. 33. Mendoza-Sanchez, R.; Corless, V. B.; Nguyen, Q. N.; Bergeron-Brlek, M.; Frost, J.; Adachi, S.; Tantillo, D. J.; Yudin, A. K. Cyclols revisited: facile synthesis of medium-sized cyclic peptides. Chem. Eur. J. 2017, 23, 13319-13322.
C3: A literature-based analysis of 13C data reveals that rigid rings from cyclic tripeptides show relatively shielded carbonyl chemical shifts (below 170 ppm), which contrasts with the relatively higher values reported for the natural product. It should be noticed that most of the reviewed data is reported in non-polar solvents while the synthetic peptide is recorded in D2O/MeOD, this should be clearly illustrated in the text. The authors justify this tendency by stating that “cyclic tripeptide can be formed only when all peptide bonds are in crown form with cis-linkage”. Even when it is true that the crown all-s-cis configuration is highly populated in cyclic tripeptides, this is reported as the exclusive conformation for cyclo-(L-Pro-L-Pro-L-Pro). Other model cyclic peptides (e.g. cyclo-(Gly-Gly-Gly) show similar population of trans-cis-cis. For the synthetic cyclo-(Gly-L-Pro-L-Glu)2), bidimensional spectra should be provided to guarantee the correct assignment of spin systems and carbonyl carbons (COSY, HSQC, HMBC, NOESY/ROESY) and the identification of key interresidual NOE contacts which contribute in determining the s-cis or s-trans arrangement of some of the amino acids (such an analysis is also described in the original paper, where s-trans conformation for the Gly-Pro linkage was identified for the isolated cyclo-(Gly-L-Pro-L-Glu)).
R3: We appreciate the reviewer’s suggestion. We have used deuterated NMR solvents to get a single peak of protons of that solvent, since it’s present in Materials and Methods Section. To emphasize the deuterated NMR solvents used in the text, the sentence in the Figure Legend Section: “Table 2. Chemical shift……..in D2O/CD3OD.”, “Figure 1. 13C NMR spectra …… in D2O/CD3OD.”, “Figure 2. 13C NMR spectra …… in D2O.”, and “Table 4. 1H NMR data …… in D2O.”
We previously have already demonstrated that the crown all-s-cis configuration is highly populated in cyclic tripeptides (Chakraborty et al. Amino acids 2015). We showed that heterotrimetric Ca atom’s angle to be either slightly larger or smaller than 60 °C with all cis-amide bonds is difficult to facilitate into cyclic tripeptides (Fig. R1, please see the attachment). For those symmetric cyclic tripeptides (cyclo(P-P-P), cyclo(MeG-MeG-MeG), cyclo(BnG-BnG-BnG) and cyclo(AllylG-AllylG-AllylG)) composed of the same three units, P, MeG, BnG and AllylG are turn-inducing units (Fig. R1A, please see the attachment). Conversely, G is not a turn-inducing unit (poor turn inducer), since cyclo(G-G-G) was unable to form from linear G-G-G directly (Fig. R1C, please see the attachment). Glycine has been rated as a turn inducer previously. We anticipated that some side chain-protected amino acids could also play a role for cyclization similar to N-substituted amino acids with lower efficiency. We found these side-chain-protected amino acids are mild turn-inducing units (Fig. R1B, please see the attachment).
To show the importance of turn-inducing unit and the related influence of side-chain-protected unit, Table R1 summarized our cyclization results and previous reported data (please see the attachment). These indicated that Boc-protected linear tripeptides were activated with pentafluorophenol for cyclization. Since our previous data can fully support the reviewer’s concern, we decided not to conduct repeated verifications but only referring our previous findings instead.
Therefore, the cyclic tripeptide’s specific structure prevents them to become an effective antibiotic as other cyclic peptides (4-14 residues) [44-48].
We added the following sentence to the “Conclusion”: “……various functionalities. The unusual antibiotic activity of natural product 1 raised our suspicion as a cyclic tripeptide. The characterization data of the reported natural product and our synthetic cyclo(GPE)2 also indicate that the reported compound may be a dimer of cyclo(GPE).”
Reference in this response:
Ref. 22. Chakraborty, S.; Lin, S. H.; Shiuan, D.; Tai, D. F. Syntheses of some a-cyclic tripeptides as potential inhibitors for HMG-CoA reductase. Amino Acids 2015, 47, 1495-1505.
Ref. 44. Lee, D.L.; Hodges, R.S. Structure-activity relationships of de novo designed cyclic antimicrobial peptides based on gramicidin S. Biopolymers 2003, 71, 28–48.
Ref. 45. Falanga, A.; Nigro, E.; De Biasi, M. G.; Daniele, A.; Morelli, G.; Galdiero, S.; Scudiero, O. Cyclic peptides as novel therapeutic microbicides: engineering of human defensin mimetics, Molecules 2017, 22, 1217.
Ref. 46. Dartois, V.; Sanchez-Quesada, J.; Cabezas, E.; Chi, E.; Dubbelde, C.; Dunn, C.; Granja, J.; Gritzen, C.; Weinberger, D.; Reza Ghadiri, M.; Parr, T. R. Jr. Systemic Antibacterial activity of novel synthetic cyclic peptides. Antimicrob. Agents Chemother. 2005, 49, 3302–3310.
Ref. 47. Scheinpflug, K.; Krylova, O.; Nikolenko, H.; Thurm, C.; Dathe, M. Evidence for a novel mechanism of antimicrobial action of a cyclic R-,W-rich hexapeptide. PLoS ONE 2015, 10, e0125056.
Ref. 48. Wiese, J.; Abelmohsen, U. R.; Motiei, A.; Humeida, U. H.; Imhoff, J. F. Bacicyclin, a new antibacterial cyclic hexapeptide from Bacillus sp. strain BC028 isolated from Mytilus edulis. Bioorg Med. Chem. Lett. 2018, 28, 558-561.
Minor Comment 1: The lower point of the present manuscript is the lack of experimental information which guarantees the reproducibility of the described work. The authors should provide supplementary material showing all the recorded spectra. Chromatograms of the pure cyclic peptides and the gradients utilized should be provided (there are some references to HPLC analysis during the manuscript but no data is shown), as well as the retention times. The method utilized for the purification of the peptides is not described in the experimental section, neither their Rf nor the purity of the final cyclic peptides. Furthermore, the authors claim to have synthesized the cyclic tripeptide cyclo-(Ser-Pro-Gly) and its dimer cyclo-(Ser-Pro-Gly)2, but no description of the synthesis is shown. This latter should be contained in the final manuscript.
Minor Response 1: We appreciate the reviewer’s suggestion. Since our previous data can fully support the reviewer’s concern, we decided not to conduct repeated verifications but only referring our previous findings instead. To meet the reviewer’s comments, we provide supplementary material showing the recorded spectra of “cyclo (GPDE(OBn)) and cyclo (GPDE).
Minor Comment 2: Finally, it should be pointed out that the quality of the English language should be improved. To some extend, ideas are not always clearly exposed and the reader faces difficulties understanding some of the statements. Sometimes prepositions and articles are omitted, reason why the revision of the language is recommended.
Minor Response 2: The original manuscript has been proofread by native speaker. Per reviewer’s suggestion, the revised manuscript has redone the proofreading to further improve its readability.

Round 2
Reviewer 1 Report
The author should demonstrate the antimicrobial acitivty of cyclo(GPE)2 to complete this research.
Author Response
We greatly appreciate the reviewer’ comment guiding us to enrich the content. The manuscript has been revised according to the reviewer’ comment and rewritten in several places to improve the readability and present it in a more coherent fashion. The revised parts of the manuscript (our responses) are represented in red, bold and underline in the text.
Comment: The author should demonstrate the antimicrobial activity of cyclo(GPE)2 to complete this research.
Response: The authors appreciate the reviewer’s suggestion. We regret that we were not more explicit in antimicrobial activity of this particular cyclic hexapeptides. First, the antimicrobial activity of peptides can now be predicted [Refs. 1 and 2]. Fernandes et al. and Arbor et al. developed the virtual library and the adaptive neuro-fuzzy inference system for prediction of antimicrobial peptides (AMPs) in helping to efficiently discover and design novel AMPs. The algorithm measures the tendency according to a Boltzmann distribution from a phase-space encompassing the structural states of random coil, α-helix, β-turn, β-sheet aggregation, and α-helix aggregation. Compared with previous publications [Refs. 1 and 2] showing peptide structure, cyclo(GPE)2 differs in regard to cyclic structure (no in vitro aggregation) and short peptide length (< 20 amino acids). Then, cyclic peptides should process similar activity to linear peptides [14, 15]. Fig. R1 shows the predicted antimicrobial activity with peptide length and in vitro aggregation (see the attachment file). The red surface areas indicate where the intersection point (green cycle) predicts that the sequence cyclo(GPE)2 is an antimicrobial peptide. However, further study may be required to provide more evidences for its bioeffects in vivo and in vitro.
Second, the potency of these antimicrobial peptides is normally not as strong as certain conventional antibiotics [22]. Third, the antimicrobial activities of cyclic hexapeptides have been known [40, 46-48]. Forth, the biological activities results of two different labs are usually not exactly matched. Therefore, we were more relying the synthesized characteristics of the individual cyclopeptides, not tentatively identified the active fractions.
Reference in this response:
Ref. 1. Fernandes, F. C.; Rigden, D. J.; Franco, O. L. Prediction of antimicrobial peptides based on the adaptive neuro-fuzzy inference system application. Biopolymer 2012, 98, 280-287.
Ref. 2. Arbor, S.; Marshall, G. R. A virtual library of constrained cyclic tetrapeptides that mimics all four side-chain orientations for over half the reverse turns in the protein data bank. J Comput. Aided Mol. Des. 2009, 23, 87-95.
Ref. 14. Davies, J. S. The cyclization of peptides and depsipeptides. J. Peptide Sci. 2003, 9, 471-501.
Ref. 15. Hamada, Y.; Shioiri, T. Recent progress of the synthetic studies of biologically active marine cyclic peptides and depsipeptides. Chem. Rev. 2005, 105, 4441–4482.
Ref. 22. Chakraborty, S.; Lin, S. H.; Shiuan, D.; Tai, D. F. Syntheses of some a-cyclic tripeptides as potential inhibitors for HMG-CoA reductase. Amino Acids 2015, 47, 1495-1505.
Ref. 40. Dahiya, R.; Gautam, H. Total synthesis and antimicrobial activity of a natural cyclohexapeptide of marine origin. Mar. Drugs 2010, 8, 2384–2394.
Ref. 46. Dartois, V.; Sanchez-Quesada, J.; Cabezas, E.; Chi, E.; Dubbelde, C.; Dunn, C.; Granja, J.; Gritzen, C.; Weinberger, D.; Reza Ghadiri, M.; Parr, T. R. Jr. Systemic Antibacterial activity of novel synthetic cyclic peptides. Antimicrob. Agents Chemother. 2005, 49, 3302–3310.
Ref. 47. Scheinpflug, K.; Krylova, O.; Nikolenko, H.; Thurm, C.; Dathe, M. Evidence for a novel mechanism of antimicrobial action of a cyclic R-,W-rich hexapeptide. PLoS ONE 2015, 10, e0125056.
Ref. 48. Wiese, J.; Abdelmohsen, U. R.; Motiei, A.; Humeida, U. H.; Imhoff, J. F. Bacicyclin, a new antibacterial cyclic hexapeptide from Bacillus sp. strain BC028 isolated from Mytilus edulis. Bioorg Med. Chem. Lett. 2018, 28, 558-561.

Reviewer 2 Report
The authors have addressed some of the recommendations pointed out in the previous report. Supplementary material has been provided, including spectroscopic data for some of the newly synthesized compounds. Additionally, some references and descriptions have been included contributing to a better understanding of the text. Even when the authors provide several elements to support the manuscript’s conclusions, there are still some critical issues to be addressed :
- The main conclusions of the manuscript rely on the proper characterization of the cyclic hexapeptide cyclo(GPE)2 (compound 2), which is in direct contradiction with previously reported data from other working groups. For this compound, all the collected spectroscopic and HPLC data should be included in the supporting material. HRMS spectra should be provided to show the isotopic pattern expected for the molecular ion and support the hexapeptidic structure. Further MS-MS experiments that show the fragmentation pattern of the molecular ion are strongly advised.
- Further bidimensional NMR experiments (i.e. TOCSY(or COSY), HSQC and HMBC) should be provided to unequivocally assign the spin systems and, particularly alpha and carbonyl carbons’ chemical shifts.
- The authors stress the idea that due to the previously described lack of antibiotic activity of similar cyclic tripeptides, as compared with cyclic hexapeptides, the reported natural product (compound 1) should be indeed a dimeric form (cyclic hexapeptide 2). If so, the bioactivity of compound 2 should be provided to confirm such an assumption, or at least it should be hypothesized - based on the mechanism of action - why the cyclic tripeptide is not expected to be active even when the amino acid sequence differs from the previously studied compounds.
- It should be commented in the manuscript that Yudin and co-workers ( Eur. J. 2017, 23, 13319 – 13322, see Supp. information) described the calculation of the chemical shifts for the reported natural cyclic tripeptide (compound 1), showing a good correlation between calculated and observed chemical shifts. This latter supports the previously proposed tripeptidic structure and must be discussed by the authors since it directly contradicts the results herein described.
- Some of the newly added descriptions lack proper concordance and should be re-written to gain clarity. For example in lines 157-160 “However, we didn’t find any cyclic tripeptide with antibiotic activity [21, 22]. So is other. In contrast, studies have disclosed the importance of ring size between 4 to 14 residues on the antimicrobial activity [44, 45]”, there is a lack of coherence among the sentences.
Even when some new information has been provided, the authors haven’t still managed to provide enough evidence to fully support their conclusions. Consequently, some critical aspects should be addressed before publication in Molecules.
Author Response
We greatly appreciate the reviewer’ comments guiding us to enrich the content. The manuscript has been revised according to the reviewer’ comments and rewritten in several places to improve the readability and present it in a more coherent fashion. The revised parts of the manuscript (our responses) are represented in red, bold and underline in the text.
Comment: The authors have addressed some of the recommendations pointed out in the previous report. Supplementary material has been provided, including spectroscopic data for some of the newly synthesized compounds. Additionally, some references and descriptions have been included contributing to a better understanding of the text. Even when the authors provide several elements to support the manuscript’s conclusions, there are still some critical issues to be addressed:
Response: The authors would like to thank the reviewer for careful and thorough reading of this manuscript.
C1: The main conclusions of the manuscript rely on the proper characterization of the cyclic hexapeptide cyclo(GPE)2 (compound 2), which is in direct contradiction with previously reported data from other working groups. For this compound, all the collected spectroscopic and HPLC data should be included in the supporting material. HRMS spectra should be provided to show the isotopic pattern expected for the molecular ion and support the hexapeptidic structure. Further MS-MS experiments that show the fragmentation pattern of the molecular ion are strongly advised.
R1: To meet the reviewer’s comments, we provide supplementary material showing the recorded spectra of “cyclo(GPDE(OBn)), cyclo(GPDE), cyclo(GPE(OBn))2, and cyclo (GPE)2, respectively”
To synthesize the cyclic tripeptide, practical aspects of solution-phase peptide synthesis are also considered with coupling protocols and fragment-based approaches for assembly of extended peptide units. Sequence analysis on specific peptides could be evaluate the extent of side reactions during synthesis and processing (e.g. in complete coupling, premature a-amino deprotection, and side chain modification) via spectrometric data. Indeed, for the unknown cyclic peptides from nature, there are some non-covalent interactions between sodium ion or other metal ion and peptides or protein, which cause a series of ion addition peaks in peptide sequencing by MS/MS. Most database search engines for protein identification do not support ion addition peaks which increase false positive results [Ref. 1]. Assembly amino acid sequence were known to generate cyclic peptides using solution-phase peptide synthesis. Therefore, MS/MS experiment is not necessary in this case.
Reference in this response:
Ref. 1. Wu, S.; Wang, J.; Liu, B.; Wang, H.; Wei, K.; Zhang, X. Yang, S. Determination of sodium ion addition sites of peptides by nanoelectrospray MS/MS sequence docking. Chin. J. Anal. Chem. 2006, 34, 1-4.
C2: Further bidimensional NMR experiments (i.e. TOCSY(or COSY), HSQC and HMBC) should be provided to unequivocally assign the spin systems and, particularly alpha and carbonyl carbons’ chemical shifts.
R2: The authors appreciate the reviewer’s suggestion. The reviewer has commented that we should have the 2D NMR method to determine for alpha and carbonyl carbons’ chemical shifts. Although we agree with the reviewer that method 2D NMR was the advanced method to identify the unknown natural peptide, since method 1H and 13C NMR was introduced by Stothers et al. (Can. J. Chem. 1964:42:1563-1576). Sequence analysis on specific peptides could be evaluate the extent of side reactions during synthesis and processing (e.g. in complete coupling, premature a-amino deprotection, and side chain modification) via spectrometric data. Besides, our cyclic peptides were synthesized by the standard solution method with confidence, and no need to verify the sequence or their chemical shifts relationships.
Reference in this response:
Stothers, J. B.; Lauterbur, P. C. C13 chemical shifts in organic carbonyl groups. Can. J. Chem. 1964, 42, 1563-1576.
C3: The authors stress the idea that due to the previously described lack of antibiotic activity of similar cyclic tripeptides, as compared with cyclic hexapeptides, the reported natural product (compound 1) should be indeed a dimeric form (cyclic hexapeptide 2). If so, the bioactivity of compound 2 should be provided to confirm such an assumption, or at least it should be hypothesized - based on the mechanism of action - why the cyclic tripeptide is not expected to be active even when the amino acid sequence differs from the previously studied compounds.
R3: The authors appreciate the reviewer’s suggestion. We regret that we were not more explicit in antimicrobial activity of cyclic hexapeptides. However, we have described the general structure frame set of cyclic tripeptides and cyclic tetrapeptide in previous studies [20, 21]. The rigidity structure of these small cyclic peptides can be compared by the easiness of their cyclization. Fig. R1 showed our cyclization results and indicated that Boc-protected linear tripeptides were activated with pentafluorophenol for cyclization (please see the attachment file). The results suggested that the rigidity structure of these small cyclic peptides is corresponded to their potential biological functions [22]. The ring frame structure of cyclic tripeptides and cyclic tetrapeptide are completely different (Fig. R2)(please see the attachment file). So are their antimicrobial activities. Since our previous data can fully support the reviewer’s concern, we decided not to conduct repeated verifications but only referring our previous findings instead.
Reference in this response:
Ref. 20. Chakraborty, S.; Tyagi, P.; Tai, D. F.; Lee, G. H.; Peng, S. M. A lead (II) 3D coordination polymer based on a marine cyclic peptide motif. Molecules 2013, 18, 4972–4985.
Ref. 21. Chakraborty, S.; Tai, D. F.; Lin, Y. C.; Chiou, T. W. Antitumor and antimicrobial activity of some cyclic tetrapeptides and tripeptides derived from marine bacteria. Mar. Drugs 2015, 13, 3029- 3045.
Ref. 22. Chakraborty, S.; Lin, S. H.; Shiuan, D.; Tai, D. F. Syntheses of some a-cyclic tripeptides as potential inhibitors for HMG-CoA reductase. Amino Acids 2015, 47, 1495-1505.
C4: It should be commented in the manuscript that Yudin and co-workers (Eur. J. 2017, 23, 13319-13322, see Supp. information) described the calculation of the chemical shifts for the reported natural cyclic tripeptide (compound 1), showing a good correlation between calculated and observed chemical shifts. This latter supports the previously proposed tripeptidic structure and must be discussed by the authors since it directly contradicts the results herein described.
R4: The authors respectively thank the reviewer’s suggestion. Synthesis of cyclo (GPE) was tried by changing the C-terminal use of proline amino acid to increase the cyclization probability (Davies, J. Peptide Sci. 2003). Three linear precursors Boc-GPE(OBn)2, Boc-PE(OBn)G and Boc-E(OBn)GP were prepared to clarify the chemical shifts for the reported natural cyclic tripeptide. Besides, cyclo (serine-proline-glycine) (SPG) and cyclo(SPG)2 were also synthesized to demonstrate the spectra of cyclic monomer and dimer containing similar amino acid sequence. Table R1 summarized the 13C chemical shift of cyclic tripeptides carbonyls (please see the attachment file). Before cyclization, the carbonyl peak of linear tripeptide is very similar to regular amide in the region of 170 to 174 ppm. But after cyclization, almost all the carbonyl peaks on the cyclic tripeptide ring are shifted to the up field lower than 170 ppm except for glutamate side chain residue. Proline (Pro) and glycine (Gly) carbonyl is always lower than 170 ppm after cyclization and the Gly-Pro linkage is less likely to have trans conformation. But carbonyl peaks reported for natural product are all above 170 ppm except Glycine 169.8 which is quite similar to the carbonyl peaks of hexapeptide cyclo(GPE)2 and cyclo(SPG)2. The results have been described in the Results and Discussion section.
Reference in this response:
Ref. 15. Davies, J. S. The cyclization of peptides and depsipeptides. J. Peptide Sci. 2003, 9, 471-501.
C5: Some of the newly added descriptions lack proper concordance and should be re-written to gain clarity. For example in lines 157-160 “However, we didn’t find any cyclic tripeptide with antibiotic activity [21, 22]. So is other. In contrast, studies have disclosed the importance of ring size between 4 to 14 residues on the antimicrobial activity [44, 45]”, there is a lack of coherence among the sentences.
R5: The authors appreciate the reviewer’s suggestion. The sentence has been rewritten as following:” ……So far, we didn’t find any cyclic tripeptide with antibiotic activity [21, 22]. We have shown that the structure-activity relationship cannot be neglected. The size of the cyclic peptide is very critical. Previous studies have disclosed the importance of ring size between 4 to 14 residues on the antimicrobial activity [44, 45] such as cyclic hexapeptides [40, 46-48].”
